# Optimizing Urban Air Pollution Detection Systems

**DOI:** 10.3390/s22134767

**Published:** 2022-06-24

**Authors:** Vladimir Shakhov, Andrei Materukhin, Olga Sokolova, Insoo Koo

**Affiliations:** 1Department of Electrical, Electronic and Computer Engineering, University of Ulsan, Ulsan 44610, Korea; 2Department of Information and Measurement Systems, Moscow State University of Geodesy and Cartography, Moscow 105064, Russia; a_materuhin@miigaik.ru; 3Institute of Computational Mathematics and Mathematical Geophysics, Novosibirsk 630090, Russia; olga@rav.sscc.ru

**Keywords:** aerosols, air pollution monitoring, mobile sensors, cumulative distribution function, system performance optimization

## Abstract

Air pollution has become a serious problem in all megacities. It is necessary to continuously monitor the state of the atmosphere, but pollution data received using fixed stations are not sufficient for an accurate assessment of the aerosol pollution level of the air. Mobility in measuring devices can significantly increase the spatiotemporal resolution of the received data. Unfortunately, the quality of readings from mobile, low-cost sensors is significantly inferior to stationary sensors. This makes it necessary to evaluate the various characteristics of monitoring systems depending on the properties of the mobile sensors used. This paper presents an approach in which the time of pollution detection is considered a random variable. To the best of our knowledge, we are the first to deduce the cumulative distribution function of the pollution detection time depending on the features of the monitoring system. The obtained distribution function makes it possible to optimize some characteristics of air pollution detection systems in a smart city.

## 1. Introduction

Due to the large increase in the number of vehicles and industrial enterprises in urban areas, real-time monitoring of the environment is necessary in order to constantly analyze the composition of the air and take measurements at the appropriate time. Great harm to human health in a megalopolis is caused by tiny floating particles, called aerosols or particulates, which pollute the atmosphere, and of course, the potential hazard of urban air is not limited to particulate air pollution. Chemical pollutants in urban air can adversely affect the nervous system, cause eye tremors, dermatitis, and other diseases that require long-term treatment, and dramatically reduce the quality of life [1].

It is necessary to continuously monitor the state of the atmosphere, process the received data in real time, and make timely decisions to reduce harmful emissions in the atmosphere. Fixed stations are designed for regular air sampling at specific locations. However, many researchers note that data from fixed stations are not sufficient for an accurate assessment of the level of aerosol pollution in the atmosphere [2,3]. An essential addition to data from fixed stations may be data received from a new generation of inexpensive, smaller, mobile, and intelligent sensor systems. The benefits of using such systems were discussed in [4]. The mobility of measuring devices can significantly increase the spatiotemporal resolution of the received data. Wireless sensor networks, including those with mobile nodes, already monitor air pollution. Mobile sensors can be carried by people (volunteers) as well as placed on vehicles. However, the quality of readings from mobile, low-cost sensors is significantly inferior to stationary sensors. Commercial low-cost sensor readings were assessed against a regulatory monitor in [5]. Based on experiments with single sensors, the authors do not recommend the use of inexpensive sensors to report absolute air pollution concentrations or to verify compliance with air quality standards. At the same time, it was mentioned that low-cost distributed sensors allow air quality trends to be monitored. Other articles were devoted to the analysis of the quality of these data [6,7], and those authors noted the great possibilities of using mobile sensors and the shortcomings of low-cost devices.

For prompt decision-making in the event of emergencies, it is necessary to obtain reliable information on the presence of pollutants. To do this, we can use both fixed stations and mobile sensors to detect air pollutants of various categories: gaseous pollutants, persistent organic pollutants, toxic heavy metals, and particulate matter. Stationary sensors almost always reliably detect the presence of pollution, but they are extremely expensive and suitable only for local monitoring. Inexpensive mobile sensors can cover the entire area, but can we be sure of timely detection of contamination? Stationary sensors can be used to test the efficiency of mobile sensors. Depending on the result, it makes sense to organize monitoring of the same area of interest with several mobile sensors.

One of the urgent problems that arise in the monitoring of megacities is to optimize a pollution detection system. This can be performed with the help of natural experiments and simulations as well as mathematical techniques that make it possible to obtain appropriate statements of optimization problems and solve them. A survey of sensor system optimization techniques [8] notes that deploying more sensor nodes will increase the overall probability of detecting events in the system, albeit at the expense of increased deployment and operational costs. In search of a compromise between the number of sensors and the latency of detection, costly full-scale and simulation experiments were used in very particular cases. Rigorous formal methods make it possible to obtain results more efficiently in the general case. Despite a number of significant advantages to the mathematical approach, there is an acute shortage of appropriate mathematical methods in the literature. In this paper, we partially fill this gap: to the best of our knowledge, we are the first to deduce the cumulative distribution function for air pollution detection time. Next, we use this result and propose a probabilistic approach to optimizing monitoring systems, taking into account the properties of the sensors used.

The problem of air pollution monitoring by mobile sensors can be considered from the standpoint of timely detection of a location (the so-called *danger zones*) where the level of air pollution exceeds a certain threshold. In other words, the mobile sensor temporarily enters a certain danger zone and detects it (Figure 1). It seems quite obvious that the time of detection by the monitoring system in a danger zone depends on both the number of mobile sensors entering the zone and the sensing quality. In this paper, the time at which detection in a danger zone takes place is considered a random variable. We present the corresponding cumulative distribution function. Two variants of the intensity of entry by mobile sensors into the danger zone are considered: when the time intervals between mobile sensor entry into the danger zone are deterministic and when the mobile sensors form a Poisson flow. The obtained distribution function makes it possible to evaluate various characteristics of the monitoring system, depending on the detection quality of the sensors used (the average time to detect a danger zone, the probability of detecting the entire danger zone within a fixed time, and so on). This function helps to formalize and solve optimization problems related to the development of a mobile air pollution detection system. This makes it possible to obtain an assessment of monitoring quality in various situations without costly testing.

We organized the rest of this paper as follows. In the next section, we describe related works. Section 3 describes low-cost sensors for measuring the presence of atmospheric aerosols. System models are presented in Section 4. Monitoring system optimization is investigated in Section 5, and Conclusions are presented in Section 6.

## 2. Related Work

Under industrialization, the large number of factories in urban areas led to a decrease in air quality and environmental degradation around the world. Air quality in a megapolis, the presence of harmful impurities, especially particulate matter, and chemical pollutants, has become an urgent problem in recent years, and many scientific studies have been devoted to this topic [9,10,11]. Traditional methods of measuring air pollution—using fixed stations or mobile laboratories—have various limitations (high cost, the impossibility of measuring in hard-to-reach places, etc.). In modern monitoring systems, air quality data are gathered using various devices (sensors, geo-sensors) connected to wireless networks. Devices distributed over a large area that are nodes of a data transmission network can collect data and transmit them to sinks or base stations. Furthermore, this information is processed in analytical centers to obtain a complete picture of the state of the atmosphere. Compared to fixed stations, a wireless sensor network provides air quality monitoring that can be more detailed. Real-time data delivery improves the accuracy of atmospheric monitoring and forecasting. Systems for monitoring urban air pollution are being developed by many scientists; descriptions of the functioning of these systems were presented in previous work. Some of these air monitoring systems include fixed stations, mobile laboratories, and mobile sensors, with the communication structure most often wireless.

A lot of work on this topic describes the problems that arise when providing wireless communication between sensors distributed in space and on an intelligent platform [12,13,14,15]. Modules in such systems are responsible for collecting and storing data, pre-processing data, and converting them into usable information. This information can be useful for making decisions in emergencies and predicting potential air pollution, depending on the time of day, wind direction, etc. The authors of many papers have noted that it is necessary to develop new methods and technologies in order to track various indicators of environmental pollution. For example, a monitoring system for assessing air pollution in Sydney uses a machine learning model that combines sparse data from fixed stations along with dense data collected from mobile sensors [13]. Another monitoring system uses sensors placed on cars for cases when the nodes’ mobility may be uncontrollable (for example, on a taxi) [14]; the authors solve two problems associated with the optimal functioning of such a mobile network: optimizing data transmission from mobile nodes, and using opportunistic communication to reduce network message transmission. In a survey of monitoring systems based on the WSN, systems were classified into three categories based on sensor carriers: the Static Sensor Network (SSN), the Community Sensor Network (CSN), and the Vehicle Sensor Network (VSN) [15].

Yet other researchers noted that a higher density of nodes in the monitored area improves monitoring [16,17,18]. Based on publications in recent years, the current trend is the use of low-cost sensors connected to wireless networks. As noted by many authors, this makes it possible to use a large number of sensors, collect real-time data from different places, and compile a detailed map of air pollution in the city [12,13,14,15,16]. In [16], the authors described an experiment to monitor gaseous air pollutants in the environment of a metropolis. A wireless network with distributed sensors to measure air pollution with a high spatial resolution was used. The measurements were carried out using devices placed at a distance of 150 m from each other. The authors noted that this network is capable of capturing high-resolution spatial and temporal changes in concentrations, but to maintain the accuracy of measurements, calibration of the sensors is necessary. The authors described a procedure for calibrating sensors to improve the functioning of the monitoring system. The authors in [17] described their research—the development of a low-cost, multi-sensor node for measuring air pollution, as well as protocols to optimize the collection of data from sensors in a WSN. An overview of state-of-art uses for low-cost sensors in environmental monitoring was presented in [18].

For air quality monitoring, it is possible to use not only stationary sensors but sensors placed on moving objects, i.e., mobile sensors. Many studies on the topic of modern monitoring in megacities are devoted to the use of special vehicle ad hoc networks (VANETs) for data collection [11,19,20,21,22]. In this case, nodes of the wireless network are sensors placed on vehicles (cars, buses) that move around the city and periodically measure the content of pollutants in the air. In [2], the authors analyzed the current state of the art, the critical problems, and the perspectives on mobile monitoring. The authors noted technological developments in recent years that can be applied to real-time air quality monitoring and that significantly improved spatial and temporal resolution of the available datasets, making air pollution maps more accurate.

The authors of [19] presented a study of real-time air pollution monitoring by sensors placed on public transport vehicles in the city of Uppsala, Sweden. The data obtained from such mobile sensors complement the measurements of stationary sensors and monitoring stations. The authors carried out experiments to assess the quality of communication and the quality of the data received. The authors of [20] described a monitoring system in which multiple sensors are placed on public transport buses. The sensors act as onboard mobile data collection centers and monitor outdoor and vehicular air quality. The novelty of this approach lies in the fact that the system provides additional sleeper nodes as part of the sensor network, thereby increasing the fault tolerance of the entire system.

The authors of [21] described their research on air monitoring in Paris using a network of mobile sensors called Pollutrack. The main purpose of the study was to determine the presence of particulate matter in the air. The measurements were carried out by devices placed on the roofs of cars. The data obtained were analyzed to determine zones with mass concentrations of pollution. The authors noted that the efficiency of using mobile sensors was higher compared to the use of known aerosol counters (for example, the Light Optical Aerosols Counter). Thanks to the large number of measurements that can be taken with mobile sensors, it has been possible to create accurate maps of areas with a high-level PM_2.5_ concentration. The authors believe such accurate maps can form the basis for requiring relevant services to respond to air pollution and make timely decisions.

Other papers described problems that arise when monitoring the atmosphere of an urban area using mobile devices [22,23,24,25,26]. For example, it is necessary to have a schedule for data collection in order to optimize the power consumption in the network. The following problems are also relevant: placement of base stations for collecting data from mobile devices; installation of sensors for use on the public transport system; correction of the received data, taking into account weather conditions. For problems where mobile data collection is preferable to using stationary sensors, you can use mobile receivers that cover the entire network of sensors and collect accumulated data from them, so the problem arises of optimal routes for mobile receivers.

To monitor air pollution with sensors, you can use various moving objects: pedestrians, cars, and public transportation. In [27], the authors described an aerial system that consists of unmanned aerial vehicles (UAVs), exhaust gas monitoring devices, and mobile control terminals. Each container is equipped with a gas collection module and sensors to determine in real-time the presence of harmful impurities in the air. The authors noted that with the help of UAVs, it is possible to obtain data for analyzing the state of the air even in hard-to-reach areas—for example, monitoring harmful emissions from ships during navigation.

Mobile sensor nodes can take measurements in different places, thereby reducing the requirement for a large number of nodes to monitor extended areas. However, for accuracy, it is necessary to collect data as often as possible, and the problem of balancing monitoring accuracy versus communication costs arises: how to set up a schedule for data transmission by mobile nodes in order to ensure monitoring quality while reducing communication costs [28,29,30]. For example, in an area where there are many vehicles with sensors, you can reduce the frequency of data transmission from some nodes to avoid possible duplication. On the other hand, in areas where the concentration of harmful substances changes dramatically, the system can be configured to increase the sensing frequency to improve monitoring accuracy.

In order to solve air pollution monitoring problems, various methods are used, including probability theory and statistical methods [30,31,32,33,34]. They are commonly used to investigate the location and identification of emission sources (for example, conditional probability functions are used for source identification). The study described in [34] presented the use of statistical methods for monitoring a territory using mobile nodes. The authors noticed that air monitoring results are unreliable due to measurement uncertainties, spatial variability, and time variations in air pollution concentrations. Thus, the timely detection of hazardous air pollution is a challenging task.

When analyzing scientific papers and patents, we did not find examples of using probability theory apparatus to evaluate various characteristics of a monitoring system related to the detection time in danger zones, taking into account the quality of the mobile sensors used. Therefore, the related optimization problems were not considered. We intend to fill this gap.

## 3. Sensors and Sensing Models

For monitoring the state of the air, the data used come from various types of sensors in environmental monitoring systems. The quality of the data obtained by these systems depends on the sensors and their parameters. The most popular types of low-cost air quality sensors are electrochemical and metal-oxide sensors, photoionization detectors, and optical particle counters [35,36]. There are many companies in the world producing inexpensive sensors for measuring the presence of atmospheric aerosols. However, the quality of the data obtained from such inexpensive sensors is often questionable. Data quality is influenced by atmospheric conditions, the concentration of the pollutants, and the time of day when measurements are taken. The factors influencing sensor measurements are described elsewhere [37,38].

The authors of [39] developed a device that can collect and transmit environmental data while moving. It was possible to increase the efficiency of collecting spatiotemporal data during the movement of the device due to a more accurate assessment of the uncertainty of spatial measurements associated with this movement.

In some work [40,41], the following model was used to estimate the concentration of polluting gas emitted by a point source of contamination:(1)$$C(x,y,z)=\frac{Q}{2\pi k_{1}}\;\;\exp\left(-\frac{y^{2}}{2g_{2}}\right)\;\;\left(\exp\left(-\frac{{\left(z-H\right)}^{2}}{g_{2}}\right)+\exp\left(-\frac{{\left(z+H\right)}^{2}}{g_{2}}\right)\right)$$
where the terms are defined as follows: 

C(x,y,z)—the pollution concentration (g/m^3^) at a point in Euclidean space (*x*, *y*, *z*), where it is assumed the source of pollution is at the origin of the coordinates;

*Q*—the level of pollutant emissions, which is considered constant; 

*H*—the effective emission height of the pollutant; and

$g_{1}$ and $g_{2}$—wind speed and atmospheric stability index.

The concentration of pollutants depends on many random factors that are difficult to account for. The higher the concentration of a substance in the air, the more likely the sensors will detect it. If the sensor is in a contaminated area for a limited time, it can be assumed that the probability of detecting contamination is a random variable. The vehicle is moving in the monitoring area at a constant speed, i.e., the operating time of the sensor is constant. Averaging over the coordinates *x*, *y*, *z*, we obtain a random value for the sensor success:
(2)p = α C(x,y,z)

Here, α is a constant that can be determined using regression analysis.

In other papers, the alternative probabilistic model was used to model sensor performance [42,43,44,45]:(3)$$p=\{\begin{array}{c} \;1,\;0\leq d\leq R\\ e^{-\beta \left(x-R\right)},\;d>R \end{array}$$
where *R* is the range over which the event is reliably detected; *d* is the average distance from the sensor to the source of pollution; and *β* is a constant depending on the characteristics of the sensor and the environment, determined experimentally or by using additional models.

Thus, further down in the text, we assume that sensors located on mobile objects detect an event to be monitored at a certain probability: *p* (entrance to an area where the level of air pollution exceeds a certain given threshold value). Highly reliable sensors can be used to estimate the likelihood of event detection by mobile sensors. We believe that the stationary sensor accurately detects the presence of aerosols in the atmosphere. These results can be used to estimate the performance of low-cost mobile sensors.

To evaluate the efficiency of detecting solid particles by low-cost mobile sensors, we apply a probabilistic approach as follows.

We introduce a discrete value, *Y*, which is 1 if the mobile sensor detects an event, but is 0, otherwise:(4)$$Y=\left\{\begin{array}{c} 1\;0\\ \;p\;1-p \end{array}\right\}$$

It is obvious that the mathematical expectation of a random variable *Y* is
(5)EY=1·p+0·(1−p)=p

It follows that to obtain the value of *p*, we can use the data obtained from stationary sensors:(6)$$p=\frac{\sum_{i=1}^{M}Y_{i}}{M}$$
where $Y_{i}$ is the *i*-th observation received by a stationary sensor, and *M* is the number of observations.

In the rest of the paper, we assume that the value of *p* is given. This value depends on the technical characteristics of the device, as well as on the concentration of pollutants in the monitoring area.

## 4. System Models

Let us consider a model for the problem of air pollution detection in some zone, *Z*, using mobile sensors (Figure 1). The mobility of sensors can be organized in different ways; here, we consider two main options. In the first case, the sensors are placed on public transport vehicles, which move according to a schedule. In the second case, the sensors move with volunteers (on foot or by car). In the first case, the time intervals between arrivals of mobile sensors in zone *Z* can be known and not random. In the second case, the intervals between arrivals of mobile sensors in zone *Z* are random. We assume this flow of mobile sensors to zone *Z* to be Poisson, i.e., the time between each input is distributed exponentially [46]. A study was conducted in Beijing, which in our opinion, can be considered an experimental confirmation of this assumption [47]. If several zones are examined, then the combined flow will also be Poisson with an intensity equal to the total intensity of the flows.

For further calculations, we need to know the exponential distribution parameter for the time between arrivals of mobile sensors in zone *Z*. To estimate the parameter, it is enough to calculate the average number of mobile sensors in zone *Z* per unit of time (intensity *λ*). Let $N_{Z}$ mobile sensors arrive in the region over a fixed time interval, *t*.

In [24], we introduced the following notations to describe the system model.

N is the total number of mobile objects (vehicles, buses, unmanned aerial vehicles, and so on), equipped with sensors for air pollution detection.

Z is a region of interest (a pollution zone).

$N_{Z}\left(t\right)$ is the number of mobile sensors that have passed through zone Z during time interval *t* for pollution detection.

τ  is a time interval between the arrivals of adjacent mobile objects. It can be random or deterministic. Both cases have been considered.

λ  is the mobile objects’ rate.

p  is the probability of detecting pollution (this property is directly related to the quality of the sensor) during the time the mobile sensor stays in zone *Z.*

*T_D_* is the pollution detection time. 

In addition, we assume that we know all the characteristics of the sensors, and we can determine the quality of their work—the ability to detect pollution. To do this, you can use preliminary experiments or special models.

First, let us consider the case when the flow of mobile objects to zone *Z* is not random, i.e., there is a deterministic time between arrivals of mobile nodes. It is logical to assume that if an event was detected in the first experiment, then the detection time will equal *τ.* Let us obtain the distribution function, *T_D_*. Pollution detection requires a random geometrically distributed number of mobile sensors. Under the definition of cumulative distribution function (CDF), we write:(7)$$F_{T_{D}}\left(t\right)=P\left(T_{D}<t\right)=p\sum_{k=1}^{k^{*}\left(t\right)}{\left(1-p\right)}^{k-1}\;$$

We use the formula for the sum of geometric progressions and obtain the CDF of detecting rate *T_D_* as follows:(8)$$F_{T_{D}}\left(t\right)=1-{\left(1-p\right)}^{k^{*}\left(t\right)-1}$$

Here,
k*(t)=argmax { k| ∀ k∈ℕ : τk<t }

Thus, *T_D_* is a continuous random variable with the following mathematical expectation, E, and dispersion, D:(9)$$E\left[T_{D}\right]=\frac{\tau }{p}$$
(10)$$D\left[T_{D}\right]=\frac{\tau^{2}\left(1-p\right)}{p^{2}}$$

Figure 2 shows a piecewise constant non-decreasing function for value *τ* = 1 and some values of *p*.

Now, consider the case of Poisson traffic flow into monitoring area *Z*; in other words, the inter-arrival time of mobile sensors, *τ*, is an independent exponentially distributed random variable. If *τ* ≥ *t*, then during time *t*, no mobile objects arrive, i.e.,
(11)$$P\left(\tau \geq t\right)=P\left(N_{z}\left(t\right)=0\right)=e^{-\lambda t}$$

Consequently, the CDF of *τ* is as follows:(12)$$F_{\tau }\left(t\right)=P\left(\tau <t\right)=1-e^{-\lambda t}$$

The average time between arrivals of mobile objects in the monitored area is known:(13)$$E\left[\tau \right]=\frac{1}{\lambda }\;$$

A sensor placed on a mobile object detects an increased level of pollution with probability *p* when entering zone *Z*. The number of mobile objects that visit *Z* from the moment of heavy pollution to its detection is a random variable, *M*, with the following geometric distribution:(14)$$P\left(M=k\right)=p{\left(1-p\right)}^{k-1},\;k=1,2,3\ldots \;$$

We calculate the detection time of critical pollution, $T_{E}$: (15)$$T_{E}=\sum_{k=1}^{\tau_{k}}\tau_{k}\;$$
where *τ_k_* denotes exponentially distributed random values with parameter λ. 

Thus, for the given *M*, the detection time has an Erlang distribution. Next, we use the law of total probability to obtain the probability density function of the random variable,$\;T_{E}$: (16)$$f_{T_{E}}\left(t\right)=\sum_{k=1}^{\infty }p{\left(1-p\right)}^{k-1}\frac{\lambda^{k}t^{k-1}}{\left(k-1\right)!}e^{-\lambda t}\;$$
and
(17)$$f_{T_{E}}\left(t\right)=p\lambda \left(\sum_{k=0}^{\infty }\frac{{\left(\lambda t\left(1-p\right)\right)}^{k}}{k!}\right)e^{-\lambda t}=p\lambda e^{\lambda t\left(1-p\right)}e^{-\lambda t}$$

Hence,
(18)$$f_{T_{E}}\left(t\right)=\theta e^{-\theta t}$$
where θ=pλ.

We have shown that $T_{E}$ has an exponential distribution with parameter *θ* and the well-known CDF.
(19)$$F_{T_{E}}\left(t\right)=1-e^{-\theta t}$$

The results obtained can be used for various applied problems, where it is necessary to manage and control the cost and efficiency of the monitoring system.

## 5. System Optimization

Let us consider some problems in optimizing the monitoring system. Air pollution in a certain area must be detected in *h* units of time with a given probability, α. Let us calculate the minimum number of sensors, *N*, installed on mobile objects.

We formulate a formal statement of the corresponding optimization problem as follows:N→min
(20)$$P\left(T_{E}<h\right)\geq \alpha \;$$

We assume the traffic rate of mobile objects linearly depends on the total number of objects involved, *λ* = *cN*, 0 < *c* < 1, where *c* is a constant. We can estimate this constant using linear regression.

In this example, *T_E_* is exponentially dependent on *pcN*. We have been given a threshold value for the contamination detection time. We calculate the probability that real-time detection of contamination does not exceed the specified value *h* with the following formula:(21)$$P\left(T_{E}<h\right)=1-e^{-pcNh}$$

The probability of detecting the air pollution level within a period not exceeding the given value *h* depends on the number of devices placed on mobile objects. Figure 3 shows this dependence. Here, it is assumed that the sensing parameter, *p*, takes values from the set {0.1; 0.3; 0.7}.

The probability, $P\left(T_{E}<h\right)$, is a monotonically increasing function of *N*. Thus,
(22)$$N_{\min}=\arg\;\left\{\;P\left(T_{E}<h\right)=\alpha \;\right\}\;$$

We can calculate the required minimal value as follows:(23)$$N_{\min}=-\frac{\ln\left(1-\alpha \right)}{pch}$$

Next, we consider the following problem of balancing the cost of a system versus its efficiency. The cost of a monitoring network depends on the number of sensors and is equal to $Na_{1}$ where $a_{1}$ is the cost of each sensor. Pollution has to be detected as soon as possible. A delay in detection leads to penalties. If $a_{2}\;$ is a delayed penalty per unit of time, then the general expenses can be defined as follows:(24)$$ET_{E}\;a_{2}+Na_{1}$$

Here $ET_{E}$ is the expected pollution detection time. Thus, using introduced designations, we obtain the following penalty function:(25)$$g\left(\lambda \right)=\frac{\lambda a_{1}}{c}+\frac{a_{2}}{p\lambda }$$

We need to minimize the penalty, so we define the optimal value of traffic intensity:(26)$$g_{\lambda }^{\prime }\left(\lambda \right)=\frac{a_{1}}{c}-\frac{a_{2}}{p\lambda^{2}}$$
and
(27)$$g_{\lambda }^{\prime }\left(\lambda^{*}\right)=0\;\Rightarrow \;\lambda^{*}=\sqrt{\frac{a_{2}c}{a_{1}p}}$$

Next,
(28)$$g_{\lambda }^{\prime \prime }\left(\lambda \right)=\frac{2a_{2}}{p\lambda^{3}}>0\;\forall \lambda >0$$

Therefore, the penalty function has a global minimum point at the value *λ**. Hence, we obtain the optimal traffic intensity of mobile objects (vehicles, pedestrians) equipped with sensors.

## 6. Discussion

We gave a rigorous mathematical foundation of the proposed original results. The formalistic conception of mathematics that has developed to date appeals to a purely logical conclusion from the initial assumptions. A mathematical derivation is valid if and only if it can be formally deduced from its premises. Once the starting points are fully formulated, everything else is built from them, without recourse to the outside world, intuition, or experiment. Thus, experimental verification can only be applied to our initial assumptions. However, this has already been performed in plenty of previous works. First, we use the assumption of probabilistic detection in the sense that a sensor node is able to detect an event (air pollution) with a certain predefined probability. This assumption is widely used in a variety of situations [48,49], including air pollution monitoring [40,50]. An experimental evaluation of the respective capabilities of the air sensors was also carried out [51,52]. Moreover, in Section 3, we discussed how to calculate this probability in a very general situation, and secondly, we assumed this flow of mobile sensors (vehicles) to the zone of interest to be Poisson. This is also a widely used principle [53] that has been experimentally confirmed in many studies [46,54,55].

## 7. Conclusions

To the best of our knowledge, the cumulative distribution function of air pollution detection time by mobile sensors has not yet been studied. We have deduced this function explicitly using the assumptions as follows. A single mobile sensor is capable of detecting air pollution with a certain predetermined probability. Arrivals of mobile sensors into the pollution zone form a Poisson process. Thus, we provide a rigorous mathematical basis for solving various applied problems related to air pollution monitoring. In particular, we use this distribution function to minimize the number of sensor nodes for a given probability of detecting pollution in a fixed time and find the optimal traffic intensity of mobile sensors. Please note that the suggested results can be applied in a wide variety of scenarios.

## Figures and Tables

**Figure 1 sensors-22-04767-f001:**
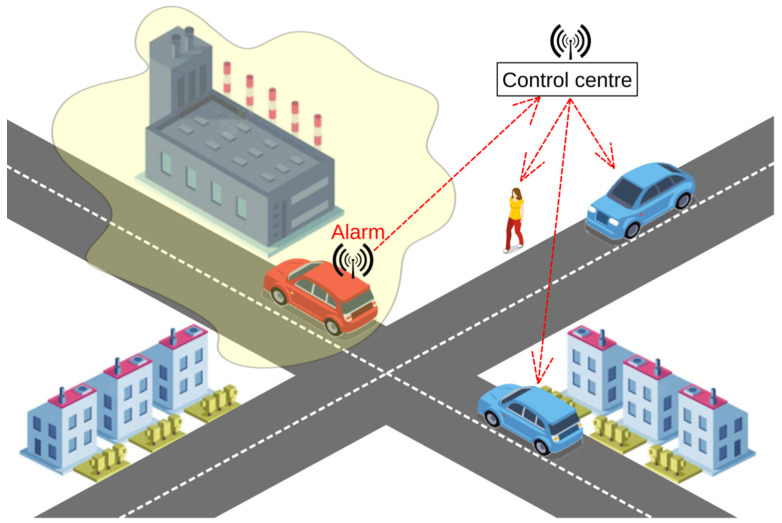
Danger zone detection.

**Figure 2 sensors-22-04767-f002:**
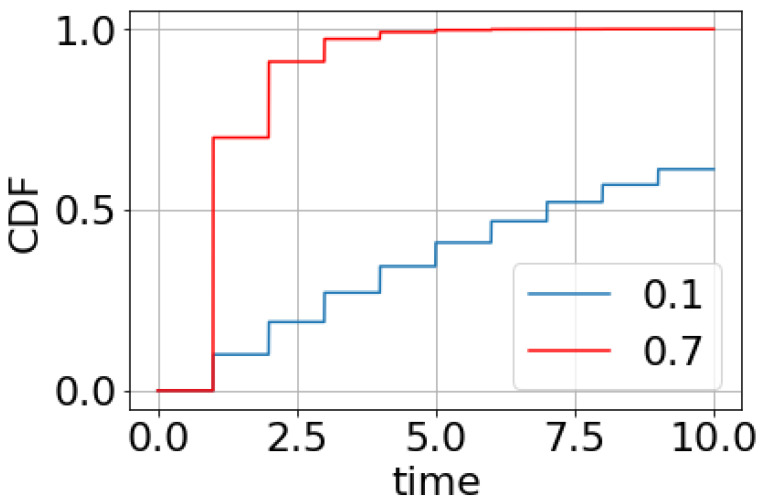
Cumulative distribution function of *T_D_* for *τ* = 1 and *p* values from 0.1 to 0.9.

**Figure 3 sensors-22-04767-f003:**
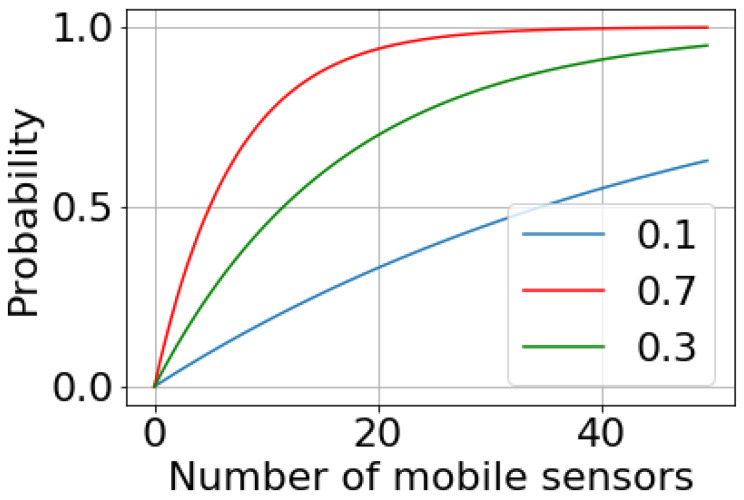
Dependence in the probability of detecting contamination within a specified time for different numbers of mobile sensors.

## Data Availability

Not applicable.

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
