# Peer review of "Optimizing Urban Air Pollution Detection Systems"

_sensors, 2022, doi:10.3390/s22134767_

Round 1

Reviewer 1 Report

I don't think the authors took into account my previous comments: you can find my previous (non-considered) comments, with some new notes.

-        Abstract: I suggest the authors to be more specific about the main results obtained from this study.

-        Paragraph "Introduction": From my point of view, this section lacks several references to the bibliography, which I consider necessary. I also suggest to the authors to review the distribution of the different concepts reported in this paragraph, as the discourse seems to me very disconnected.

>>> I do not think that the authors have deepened the scientific literature. On the contrary, it seems to have been reduced compared to the first version of the manuscript.

>>> The introduction still seems quite disjointed to me, with similar concepts repeated several times.

-        Paragraph "Introduction": I suggest to specify the purpose of the work, in the best possible way.

>>> The authors report “Despite a number of significant advantages to the mathematical approach, there is an acute shortage of appropriate mathematical methods in the literature. In this article, we partially fill this gap. ”: Really suggested to better specify the aim of the work.

-        Paragraph "Conclusions": I do not understand how the authors can arrive at these conclusions, in a solid way, based only on what is reported in the text. In this paragraph I suggest reporting the limitations of the study / approach used.

>>> I really don't think this paragraph has improved in any way.

-        Line 74 (but in general in the text): Pay attention to the subscript of PM10.

>>> Attention to line 163.

Author Response

Comment 1

Reviewer’s

Comment:

Abstract: I suggest the authors to be more specific about the main results obtained from this study.

Authors’ Reply:

To reflect the reviewer’s comment, we highlights the main result in Introduction.

Comment 2

Reviewer’s

Comment:

Paragraph "Introduction": From my point of view, this section lacks several references to the bibliography, which I consider necessary. I also suggest to the authors to review the distribution of the different concepts reported in this paragraph, as the discourse seems to me very disconnected.

Authors’ Reply:

To reflect the reviewer’s comment, we modified Introduction. Unfortunately, the reviewer did not mentioned specific necessary references to the bibliography and concepts. If items will be provided, we will process them.

 Comment 3

Reviewer’s

Comment:

Paragraph "Introduction": I suggest to specify the purpose of the work, in the best possible way. Paragraph "Conclusions": I do not understand how the authors can arrive at these conclusions, in a solid way, based only on what is reported in the text. In this paragraph I suggest reporting the limitations of the study / approach used.

Authors’ Reply:

To reflect the reviewer’s comment, we modified Introduction and Conclusion.

Comment 4

Reviewer’s

Comment:

Line 74 (but in general in the text): Pay attention to the subscript of PM10.

Attention to line 163.

Authors’ Reply:

To reflect the reviewer’s comment, we fixed this items.

Reviewer 2 Report

This manuscript is focusing on the actual problem - urban air pollution. The authors rightly point out that air pollution data obtained from fixed stations are insufficient for an accurate assessment of the level of pollution. It is necessary to use mobile measuring devices that can improve the quality of monitoring, and therefore the responsiveness of the relevant services to troubleshoot problems. In the text, the authors describe related works on the topic of monitoring systems using mobile sensors. They indicate the need to evaluate the characteristics of systems depending on some data collection conditions. The authors have derived the form of the cumulative distribution function of the pollution detection time depending on the features of the monitoring system.

Certainly, the presented result can optimize the functioning of some pollution detection systems and smart city applications. This is a solid work and a well-written article. I recommend to accept this one, however, there is a mirror note to authors to improve readability of the text as follows. The paper does not contain a visual picture for a simple perception of the problem statement and the environment associated with it. An appropriate picture will improve the readability of the paper.

Author Response

To reflect the reviewer’s comment, we created and added a picture.

Reviewer 3 Report

The issues addressed in this manuscript are of current global environmental importance. The proposed method solves practical problems in the field. However, there is one point that the author needs to elaborate on. As far as possible to supplement the specific experimental verification part, to prove the consistency of theory and practice. Among them, it is necessary to complete the construction of experimental system and data collection, and evaluate the feasibility.

Author Response

To reflect the reviewer’s comment, we created a new section: Discussion.

Reviewer 4 Report

With their contribution, the authors address an important topic of our time. Due to increasing industrial activities and increasing vehicle traffic, air pollution is increasing sharply. Extensive monitoring with locally resolved information is therefore desirable.

In the paper, the authors focus on particulate matter pollution in the air. But Urban Air Pollution is much more than this (also includes chemical pollutants in the air). Therefore, the title of the paper is somewhat misleading and should be changed according to the chosen topic.

The authors state that "the use of these mobile devices is associated with a decrease in the quality of the information". Unfortunately, there are no explanatory descriptions of what is meant by that. Elsewhere (line 51) they write: "the quality of pollution detection may not be quite good enough". This is a very general statement. The reviewer misses a definition of good and bad air pollution detection. What is the goal? What "quality" in determining air pollution is desired?

The approach chosen by the authors is new, the description of the underlying mathematical models is good. For the reviewer, the question arises as to the size of zone Z in which air pollution detection is to be carried out. In line 302 the authors define p as "the probability of detecting pollution during the time the mobile sensor stays in zone Z". They define this as "sensor quality". This term in misleading and should be replaced throughout the text. Sensor quality includes much more details.

The main shortcoming of the present paper is the complete lack of experimental confirmation and comparison of the results with the state of the art. What is the possible spatial resolution of the measured values? How big is the relevance of a high local resolution? How do results improve with the authors' approach compared to previous approaches?

Author Response

Comment 1

Reviewer’s

Comment:

In the paper, the authors focus on particulate matter pollution in the air. But Urban Air Pollution is much more than this (also includes chemical pollutants in the air).

Authors’ Reply:

Since the proposed results can be applied for any types of pollutants, we have made the corresponding adjustments to the paper.

Comment 2

Reviewer’s

Comment:

The authors state that "the use of these mobile devices is associated with a decrease in the quality of the information". Unfortunately, there are no explanatory descriptions of what is meant by that. Elsewhere (line 51) they write: "the quality of pollution detection may not be quite good enough". This is a very general statement. The reviewer misses a definition of good and bad air pollution detection. What is the goal? What "quality" in determining air pollution is desired?

Authors’ Reply:

To reflect the reviewer’s comment, we have reformulated all of these points.

Comment 3

Reviewer’s

Comment:

The approach chosen by the authors is new, the description of the underlying mathematical models is good. For the reviewer, the question arises as to the size of zone Z in which air pollution detection is to be carried out. In line 302 the authors define p as "the probability of detecting pollution during the time the mobile sensor stays in zone Z". They define this as "sensor quality". This term in misleading and should be replaced throughout the text. Sensor quality includes much more details.

Authors’ Reply:

In this article, we do not utilize the size of zone Z in any way, while assuming that the sensor is able to detect the event of interest with some pre-calculated probability. This is widespread in the literature. To reflect the reviewer’s comment, we have added the corresponding references. Also, we replaced the term "sensor quality".

Comment 4

Reviewer’s

Comment:

The main shortcoming of the present paper is the complete lack of experimental confirmation and comparison of the results with the state of the art. What is the possible spatial resolution of the measured values? How big is the relevance of a high local resolution? How do results improve with the authors' approach compared to previous approaches?

Authors’ Reply:

To reflect the reviewer’s comment, we created the new section: Discussion.

Round 2

Reviewer 1 Report

-